# Long-Term Outcomes of Lymph Node Transfer in Secondary Lymphedema and Its Correlation with Flap Characteristics

**DOI:** 10.3390/cancers13246198

**Published:** 2021-12-09

**Authors:** Dimitrios Dionyssiou, Alexandros Sarafis, Antonios Tsimponis, Asterios Kalaitzoglou, Georgios Arsos, Efterpi Demiri

**Affiliations:** 1Department of Plastic Surgery, Papageorgiou General Hospital, School of Medicine, Faculty of Health Sciences, Aristotle University of Thessaloniki, 56403 Thessaloniki, Greece; alexsarafis@gmail.com (A.S.); atsimponis@gmail.com (A.T.); depydemiri@gmail.com (E.D.); 23rd Department of Nuclear Medicine, Papageorgiou General Hospital, School of Medicine, Faculty of Health Sciences, Aristotle University of Thessaloniki, 56403 Thessaloniki, Greece; kalaitzoglou.a@gmail.com (A.K.); garsos@auth.gr (G.A.)

**Keywords:** breast cancer, mastectomy, lymphedema, lymph node transfer, lymph node flap

## Abstract

**Simple Summary:**

Upper limb lymphedema is a common complication following breast cancer treatment. Vascularized Lymph Node Transfer (VLNT) is an emerging therapeutic modality with satisfactory outcomes. However, surgical complexity and potential donor-site morbidity may complicate its application. The aim of our retrospective study was to assess the impact of certain lymph node flap characteristics on long-term outcomes. In a series of 64 post-mastectomy lymphedema patients who underwent VLNT with the Selected Lymph Node technique, we confirmed a positive correlation between lymph node flap size and number of lymph nodes transferred, as well as between flap size and achieved lymphedematic volume reduction. Lymphedema stage and flap vascular pedicle had no significant impact on the final results. These findings underline the necessity for meticulous flap choice, in order to combine a flap harvest of adequate size with more favorable outcomes and minimized donor site morbidity.

**Abstract:**

Background: This retrospective study aimed to assess the impact of certain flap characteristics on long-term outcomes following microsurgical treatment in Breast Cancer-Related Lymphedema (BCRL) patients. Methods: Sixty-four out of 65 BCRL patients, guided by the “Selected Lymph Node” (“SeLyN”) technique, underwent Vascularized Lymph Node Transfer (VLNT) between 2012 and 2018. According to their surface size, flaps were divided into small (<25 cm^2^, *n* = 32) and large (>25 cm^2^, *n* = 32). Twelve large and six small flaps were combined with free abdominally based breast reconstruction procedures. Lymphedema stage, flap size, vascular pedicle and number of lymph nodes (LNs) were analyzed in correlation with long-term Volume Differential Reduction (VDR). Results: At 36-month follow-up, no major complication was recorded in 64 cases; one flap failure was excluded from the study. Mean flap size was 27.4 cm^2^, mean LNs/flap 3.3 and mean VDR 55.7%. Small and large flaps had 2.8 vs. 3.8 LNs/flap (*p* = 0.001), resulting in 49.6% vs. 61.8% VDR (*p* = 0.032), respectively. Lymphedema stage and vascular pedicle (SIEA or SCIA/SCIP) had no significant impact on VDR. Conclusion: In our series, larger flaps included a higher number of functional LNs, directly associated with better outcomes as quantified by improved VDR.

## 1. Introduction

Lymphedema occurs as a sequence of lymphatic system insufficiency and impaired lymphatic drainage [1]. Deranged lymphatic flow results from either congenital or acquired abnormalities of lymphatic transport, leading to lymph-fluid accumulation in the interstitial space [2]. It can be primary, as a consequence of intrinsic developmental lymphatic system failure, or secondary, when a specific cause such as surgical intervention, malignancy, trauma, radiation, infection, inflammation, etc., can be identified [3].

Upper limb lymphedema, as an iatrogenic complication or side effect of breast cancer treatment, is a potentially devastating medical condition, commonly referred to as Breast-Cancer-Related Lymphedema (BCRL), showing a prevalence ranging from 4 to 49 percent [4]. Severe damage or destruction of the upper limb lymphatic pathways by lymphadenectomy, radiotherapy, chemotherapy, or combination treatment are the main causes of BCRL, whereas it is only rarely attributable to the infiltration of lymphatics by the cancer itself [5]. BCRL, especially in more advanced clinical stages, is characterized by notable physical and psychological morbidity. It, moreover, incurs considerably high medical costs, with a crucial impact on National Health System economics [6].

Even though conservative treatment serves as the keystone for the symptomatic improvement of lymphedema and may cease or decelerate the progression of the disease, a combined approach of conservative therapy and surgical intervention aspires to achieve more optimal results [7]. Vascularized Lymph Node Transfer (VLNT) is an emerging microsurgical treatment modality, considered to be a physiologic procedure that provides satisfactory outcomes, especially when applied at early stages or in mild lymphedema cases [8,9]. Following scar tissue release, microsurgical lymph node transplantation offers healthy vascularized lymphatic tissue, which promotes lymphangiogenesis through the production of Vascular Endothelial Growth Factor C, improves the immunologic functions of the affected limb, and may functionally bridge the destructed lymphatic channels [7].

Thus far, VLNT has shown promising outcomes; however, surgical complexity and potential donor-site morbidity may complicate its application [10]. Amongst multiple donor sites proposed, the inguinal area remains popular. It offers sufficient lymph node tissue, ease of flap harvest, and reproducibility of surgical approach based on anatomical landmarks, allows aesthetic closure of the donor scar into the groin area, and ensures minor donor-site morbidity. A review of the literature certifies its safety as a surgical intervention; it still bears, however, a 1.6% risk of iatrogenic donor-site lymphedema [11].

A recent study introduced a new technique, the “Selected Lymph Node” (“SeLyN”) technique, which identifies the most suitable functional lymph nodes (LNs) of the inguinal area guided by a SPECT-CT radiocolloid lymphoscintigraphy. In this study, it was concluded that “SeLyN” is an effective and safe technique, with a shortened overall operating time, yielding improved outcomes and reduced donor-site morbidity [12].

Despite the fact that numerous studies assert the efficacy of VLNT in BCRL patients [7,9,10,12,13,14], the unpredictable favorable outcome and the exact functional mechanism remain a matter of debate [13]. In terms of lymphatic reconstruction, many aspects of these microsurgical procedures and their precise role in lymphedema treatment are yet to be elucidated. 

We hypothesized that larger flaps will carry more lymph nodes and that more lymph nodes will provide favorable treatment outcomes. Therefore, the present study aimed to evaluate certain VLNT flap characteristics in relation to their impact on long-term outcomes for BCRL patients. 

## 2. Materials and Methods

### 2.1. Patients

A retrospective cross-sectional study was conducted on upper extremity BCRL patients who underwent functional lymphatic reconstruction with the “SeLyN” technique from January 2012 to December 2018. All female patients at the age of 18 years old or above, suffering from upper limb BCRL stage I, II, or III according to the Staging System of the International Society of Lymphology [1], were eligible to be included in the study. All patients could be candidates for a lymph node transfer operation, unless they were being subjected to an active cancer or metastatic disease treatment, were diagnosed with a bilateral upper limb lymphedema, or had a history of bilateral breast cancer, history of primary lymphedema, limb edemas of different etiologies, coagulopathy, pregnancy, body mass index above 35 kg/m^2^, alcohol or drug abuse, or were unable to comply with the proposed treatment and follow-up protocol [12]. Sixty-four out of 65 consecutive female patients were finally included in the study.

Lymphedema diagnosis was based on detailed clinical examination, followed by non-contrast Magnetic Resonance (MR) lymphography, indocyanine green fluoroscopy, and ^99m^Tc-nanocolloid lymphoscintigraphy. Eleven patients were classified as Stage I, thirty-four as Stage II, and nineteen as Stage III lymphedema, according to the International Society of Lymphology (ISL) staging system for lymphedema [1].

All patients had undergone a functional lymphatic reconstruction with microvascular lymph node transplantation [9]; patients unable or unwilling to comply with the proposed treatment protocol, as well as women with a history of primary lymphedema or bilateral or metastatic breast cancer, were excluded. The study was conducted according to the guidelines of the Declaration of Helsinki and approved by the Institutional Review Board of “Papageorgiou” General Hospital, Thessaloniki, Greece (protocol code 526 and date of approval 15 November 2018). Written informed consent was obtained from all participants.

### 2.2. Treatment Protocol, Surgical Technique, and Management

Conservative treatment modalities for lymphedema, such as Complex Decongestive Physiotherapy (CDP), body mass index (BMI) improvement, arm elevation, and antibiotic therapy in cases of infection episodes, were all undertaken before proceeding to VLNT.

Patients’ demographics, lymphedema stage, and etiology were preoperatively recorded. Volumetric measurements were made in both upper limbs using the truncated cone formula based on 4 cm intervals as the serial perimeter [15]; the amount of volume excess was calculated as the differential ratio between the edematous and the unaffected upper extremity, termed Volume Differential (VD): [(Affected limb volume−Unaffected limb volume)/Unaffected limb volume] × 100 = VD% [16]. Patients with preoperative VD less than 10% were excluded from the study.

Bilateral lower limb ^99m^Tc-nanocolloid planar and SPECT-CT lymphoscintigraphy was preoperatively performed, to map the superficial inguinal LNs and select the groin related to the limb of the best lymphatic drainage. The “SeLyN” technique protocol [12] was meticulously applied in all patients’ inguinal areas, and a handheld Doppler indicated which of the superficial inferior epigastric artery (SIEA), the superficial circumflex iliac artery (SCIA), or its perforator (SCIP) represented the dominant pedicle of the selected flap.

The same surgical technique was used in all patients, starting with fibrotic tissue release and clearance of the axilla, until a well-vascularized bed was identified, followed by preparation of a vascular pedicle of adequate length at the recipient site; vessels of choice were either branches of the thoracodorsal or the lateral thoracic. Intraoperatively, flap consistency with the preoperative measurements was documented, as well as the size of each flap, the number of LNs transferred within the flap, along with the supplying vessels of the flap pedicle. The number of LNs situated in the flap was evaluated by palpation, intraoperative ultrasound examination, and indocyanine green fluoroscopy. The flap was then fully raised and transferred as a free vascularized lymph node flap to the axillary site, where it was microsurgically anastomosed to the previously prepared vessels. In cases of synchronous breast reconstruction with a deep inferior epigastric perforator (DIEP) flap, there was always a separate anastomosis performed for the LN flap at the axilla, in addition to the main DIEP flap anastomosis at its recipient site.

Postoperatively, all patients continued a manual lymphatic drainage (MLD) for 30 days, followed by the use of pressure garments of 20 mmHg for a 5-month period. Twelve months after the intervention, lymphoscintigraphy and/or ICG fluoroscopy of the operated upper extremities as well as of the donor sites’ lower limbs were performed, in order to evaluate the functionality and viability of the transferred LNs, as well as the integrity of the lymphatic circulation of the lower limb. Patients were then followed up yearly, recording upper limb volumetric changes and donor-site complications.

In order to evaluate patients postoperatively and determine their response to the applied intervention, Volume Differential Reduction (VDR) was calculated, as follows: [(Preoperative VD−Postoperative VD)/Preoperative VD] × 100 = VDR% [16]. All pre- and postoperative measurements were made constantly at morning time.

Correlation analysis between lymphedema stage, flap size, number of LNs, and vascular pedicle was used to determine each individual role in volume reduction in a long-term period; all measurements were recorded at preoperative evaluation and at three years postoperative follow-up. Flaps were equally divided into two groups, measuring the surface size by the end of harvest—Group A (small flaps) and Group B (large flaps)—and final outcomes were examined.

### 2.3. Statistical Analysis

Statistical analysis was conducted through the SPSS (edition 23) software package. Kolmogorov–Smirnov test was used to check continuous variables for normality. The non-parametric Mann–Whitney U test was used to determine the statistically significant differences in the number of LNs or Volume Differential Reduction (VDR) between Groups A and B, as well as in different stages of lymphedema and main supplying vessels (SCIA/SCIP or SIEA). The Spearman and Pearson correlation coefficient was used to assess the correlation between flap size and Volume Differential Reduction. The correlation between flap size and number of LNs was assessed by means of logistic regression. The level of statistical significance was determined as a *p* value < 0.05.

## 3. Results

### 3.1. Patients

Sixty-five female patients, with a mean age of 49.8 years (ranged 32 to 76 years) and an average body mass index (BMI) of 28.5 kg/m^2^ (ranged 20 to 35 kg/m^2^), received inguinal VLNT for BCRL, during the seven years of the study. 

The postoperative period was uneventful for 64 of the study patients. One patient had an injury at her operated arm on the first postoperative day, which caused pedicle extirpation and flap failure; the patient was excluded from any analysis during the study. No other major flap or donor-site complication was recorded, including the absence of donor-site lymphedema, which was evaluated clinically and by ICG lymphography or lymphoscintigraphy. Final follow-up measurement for this study was performed at 3 years after the surgery for each included patient. 

### 3.2. Lymph Node Flap Characteristics—Correlation with Outcomes

The average flap length was 6.1 cm (4 to 7.8 cm) and average width was 4.6 cm (3.1 to 6.8 cm); the palpated thickness of each flap never surpassed 1 cm, with minimal variation between different flaps, and was thus not taken into consideration in terms of estimating their size in cubic centimeters (cm^3^). The mean flap harvested surface size was calculated at 27.4 cm^2^, ranging between 13.0 and 47.5 cm^2^. Having divided the number of flaps equally in half, the smallest 32 flaps constituted Group A, while the largest 32 flaps constituted Group B, and the cut-off flap size was indicated at 25 cm^2^ (Table 1 and Table 2). The mean flap size was 19.8 (±3.5) cm^2^ (range 13.0 to 24.5 cm^2^) in Group A (Figure 1) and 35.0 (±6.3) cm^2^ (range 25.2 to 47.5 cm^2^) in Group B (Figure 2).

Eighteen of the 64 harvested VLNT flaps (28%) were combined with DIEP flaps. Six out of the 32 small flaps (19%) and 12 out of the 32 large flaps (38%) were such cases.

Each of the inguinal flaps contained two to five lymph nodes, with a mean number of 3.3 LNs transferred per patient. Group A contained a mean of 2.8 (±0.9) LNs, whereas Group B had a mean of 3.8 (±0.7) LNs (*p* = 0.001). There was a statistically significant positive correlation between flap size and number of LNs (*p* < 0.001, r = 0.668). This observation was more obvious in very small and very large flaps; sizes smaller than or equal to 20 cm^2^ (*n* = 13) contained an average of 2.4 LNs, while flaps larger than 35 cm^2^ (*n* = 17) accommodated an average of four LNs.

Upper limb perimeter measurements were decreased in all cases; this was directly reflected in the lymphedematous limb postoperative volume reduction. More specifically, the mean Volume Differential Reduction (mVDR) was measured at 55.7%. Group A resulted in a 49.6 (±30)% mVDR, while Group B large flaps lead to 61.8 (±9.7)% mVDR (*p* = 0.032). The correlation between flap size and volume differential reduction was also found to be positive (r = 0.312) and statistically significant (*p* = 0.012) (Figure 3). 

The correlation of lymph node number per flap and Volume Differential Reduction was also investigated and found to be positive (r = 0.370) and statistically significant (*p* = 0.003).

Moreover, mVDR was calculated at 54.4% for Stage I (*n* = 11), 56.2% for Stage II (*n* = 34), and 61.2% for Stage III patients (*n* = 19), showing no statistically significant difference between lymphedema stages: *p*(I&II) = 0.41, *p*(I&III) = 0.16, *p*(II&III) = 1.00.

Concerning the vascular pedicle, the SCIA or the SCIP was identified as the dominant vascular axis in 38 patients, while the remaining 26 of the study flaps were based on the SIEA. The SCIA/SCIP flaps had a mean size of 27.3 cm^2^, carrying an average of 3.3 LNs and resulting in 55.6% mVDR, while the SIEA flaps group had a mean size of 27.5 cm^2^, with 3.2 LNs and 55.8% mVDR accordingly. The improvement was equally distributed between SCIA/SCIP and SIEA flaps, yielding no significant impact of the supplying vessels’ identity on volume reduction (*p* = 0.967).

Although it was not a subject of the present study, all patients reported subjective postoperative improvements in their symptoms.

## 4. Discussion

Vascularized Lymph Node Transfer is an effective therapeutic modality for post-mastectomy lymphedema patients. In recent years, VLNT results have been proven to be very promising; these early results, however, have yet to be fully understood and evaluated [17,18,19]. While numerous articles describe the successful use of VLNT for lymphedema patients [7,8,9,10,12,13,14,17], there is a small number of articles in the literature regarding VLNT flap characteristics that might influence or even favor its final outcomes. 

Comparable satisfactory outcomes to previously published VLNT studies [9,10,11,12,13,14] were also achieved in the present study, after lymphatic microsurgery in BCRL patients. A long-term postoperative improvement in the circumferential measurements advocated for a significant volume reduction in terms of Volume Differential Reduction (mVDR = 55.7%). Regarding different lymphedema stages, no statistically significant difference was found in measured mVDR. 

In order to assess certain VLNT flap characteristics, flap size was examined first, in correlation with the noted volume reduction (mVDR). According to Patel et al. [20], the transfer of greater lymph node flaps is likely to better improve the lymphatic clearance in the affected upper limb. Based on this observation, the flaps of the present study were divided into large and small, with a threshold at 25 cm^2^. Mathematical analysis revealed a statistically significant positive correlation between flap size and volume reduction (49.6% mVDR for small flaps compared to 61.8% mVDR for large flaps).

The “flap size–volume reduction” correlation could be partially attributed to the inclusion of a greater number of LNs and higher lymphatic tissue densities in larger flaps [20]. In agreement with this hypothesis, a statistically significant positive correlation between flap size and number of transferred LNs was found in our series. The mean number of LNs per flap in the present study was 3.3, with an average of 3.8 LNs for flaps > 25 cm^2^ and 2.8 LNs for flaps < 25 cm^2^. Furthermore, a statistically significant positive correlation was revealed between the number of LNs per flap and volume reduction.

Cheng et al. underline preservation of the LNs and the soft tissue with the vascularized groin lymph node flap as the key to successful outcomes [21]. They, moreover, suggest, that, although the mean number of axillary LNs removed (26.8 ± 10.8) was much higher than the mean transferred superficial groin LNs (6.2 ± 1.3), the latter seem to be sufficient to gradually drain the excessive lymph into the venous system, when transplanted to the lymphedematous upper extremity [21]. In the present study, two to five LNs were included in each flap and led to positive outcomes for the majority of the operated BCRL patients. However, a greater average number of transferred LNs within larger flaps was directly correlated with improved mVDR, compared to the small flaps group with slightly lower volume reduction rates.

Groin lymph node flap has gained popularity amongst surgeons treating upper limb lymphedema, mainly due to its reliable vascular anatomy and lymph node quantity [20]. In spite of these favorable flap characteristics, special consideration is still needed, bearing in mind the potential for iatrogenic lower extremity lymphedema, even if the amount of lymph nodes is minimal [11,22,23,24]. It is appropriately expected that the removal of more LNs from the patient’s inguinal basin bears an increased risk of donor-site morbidity. Thus, the present study’s finding that larger flaps carrying more LNs offer improved lymphedema outcomes highlights the necessity for careful selection, meticulous harvest, and safe transfer, in order to maintain minimal donor-site morbidity rates. Applying the “SeLyN” technique seems to minimize donor-site morbidity and, thus far, eliminate the risk for lower limb iatrogenic lymphedema [12]. It moreover ensures selection of the most functional (radioactive) lymph node group, an additional characteristic that contributes to optimized results [12]. All patients of the present study were operated by the aforementioned technique. No lower limb lymphedema or other major donor area complication was recorded. 

There are only a few studies examining the issue of the dominant vascular pedicle in VLNT [9,12,25], while most of the authors prefer the SCIA as the safest vascular axis for microlymphatic reconstruction [20,26,27]. In our surgical practice, we agree with Gharb et al. [27] and the unpublished data of Corinne Becker (D.D. personal communication) [28], according to which the lymph node flap harvest should be based on a perforator-based approach. This allows for the evaluation of intraoperative findings and compliance with preoperative planning, by the direct visual selection of the dominant vascular pedicle, together with the number of transferred LNs. The combination of this approach with the “SeLyN” technique offers the advantage of the intraoperative choice of the largest and best situated pedicle into the most functional lymph node group available for safe harvest [12]. The present study concludes that amongst 64 cases of VLNT for BCRL, the SCIA and the SCIP were the supplying vessels in 38 patients, while the remaining 26 of the study flaps were based on the SIEA. Approximately all groin vascular pedicles were characterized by similar statistically superior outcomes. This last result reinforces the arguments in favor of our strategy, for the intraoperative direct visual selection of the dominant pedicle.

Saaristo A.M. et al. showed that the utilization of VLNT, as part of a combined breast microvascular reconstruction, aims to improve lymphedema [29]. Chang E.I. et al. demonstrated the efficacy and safety of combining DIEP flaps with lymph node transfer, which can potentially improve patients’ quality of life [30]. Likewise, Forte et al., with their systematic review concerning combined VLNT with DIEP or TRAM breast reconstruction procedures, concluded that the majority of favorable breast reconstruction results were combined with a reduction in the circumferential size of the affected upper limb, in addition to a reduction in infectious intercurrences such as cellulitis [31]. Eighteen of our cases were synchronous breast and BCRL reconstructions, using DIEP and VLNT flaps. It is obvious that the harvest of groin lymph node flaps simultaneously with the abdominal free flaps allowed the transfer of larger flaps (12 out of the 18 combined cases had LN flaps larger than 25 cm^2^), offering more favorable results, within the scope of the positive “flap size–volume reduction” correlation. 

Acknowledging the fact that the main limitation of this study is its retrospective nature, we also report the complexity of the evaluation the multifactorial parameters of various patients’ characteristics, including body mass index, age, medical status, medications, diverse characteristics of same lymphedema stage, etc., in combination with the degree of lymphedema amelioration according to volume reduction and/or infection episode reduction and/or quality of life improvement.

## 5. Conclusions

As the field of microlymphatic surgery continues to expand, understanding certain flap characteristics, such as flap size, number of lymph nodes, and dominant pedicle identity, will improve the decision-making process related to surgical planning. Flap size was directly correlated with better outcomes in the present study. We found that large flaps (>25 cm^2^) carry more lymph nodes and lymphatic tissue, which leads to an improved volume reduction in the lymphedematous limb. Finally, synchronous breast reconstruction with DIEP flaps proved to be advantageous for Breast-Cancer-Related Lymphedema treatment, allowing the transfer of larger lymph node flaps and offering more favorable results. Our findings underline the necessity for meticulous flap choice, in order to combine a flap harvest of adequate size with more favorable outcomes and minimized donor-site morbidity. The “Selected Lymph Node” (“SeLyN”) technique, based on SPECT/CT lymphoscintigraphy, ensures the safety and effectiveness of the procedure.

## Figures and Tables

**Figure 1 cancers-13-06198-f001:**
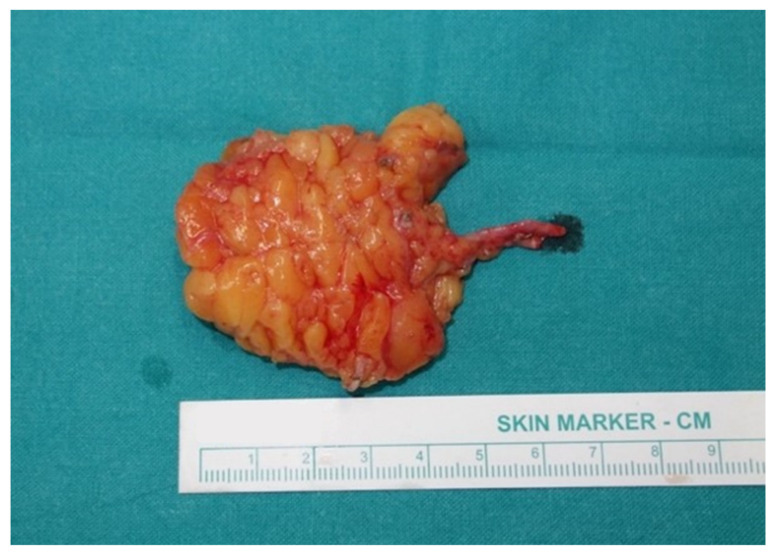
Small lymph node flap, surface size 20.2 cm^2^, including two lymph nodes.

**Figure 2 cancers-13-06198-f002:**
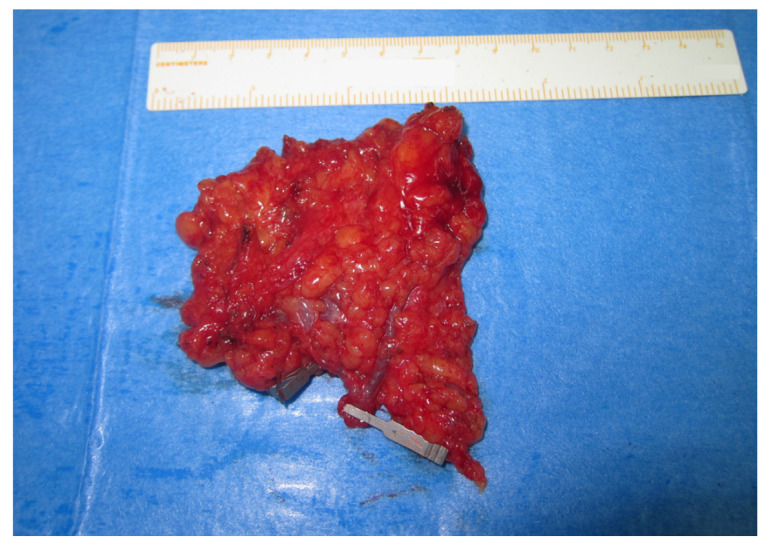
Large lymph node flap, surface size 40.6 cm^2^, including four lymph nodes.

**Figure 3 cancers-13-06198-f003:**
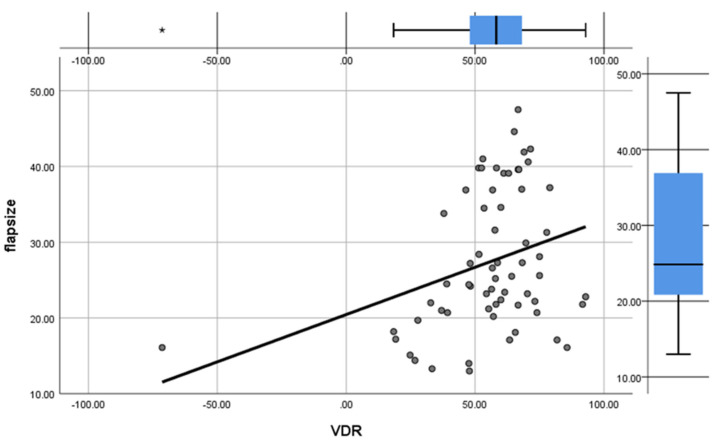
Scatter plot showing positive correlation between flap size and Volume Differential Reduction (VDR). * indicates an outlier value for VDR.

**Table 1 cancers-13-06198-t001:** Group A: Small lymph node flaps (<25 cm^2^).

Flap Size (cm^2^)	Number of LNs	Harvested Flap	Vascular Pedicle	Lymphedema Stage	Preop VD%	Postop VD%	VDR
13.0	2	LNT	SIEA	2	23	12	48
13.3	2	LNT	SIEA	2	30	20	33
14.0	2	LNT	SCIA	1	15	8	47
14.4	2	LNT	SCIA	2	30	22	27
15.1	2	LNT + FALD	SCIA	1	16	12	25
16.1	3	LNT	SCIA	2	35	5	86
16.1	2	LNT + DIEP	SCIA	1	14	24	−71
17.1	3	LNT	SCIP	1	11	2	82
17.1	3	LNT	SCIA	2	41	15	63
17.2	2	LNT	SIEA	1	26	21	19
18.1	3	LNT	SCIP	2	93	32	66
18.2	2	LNT	SCIP	2	27	22	18
19.7	3	LNT	SCIP	2	25	18	28
20.2	2	LNT	SIEA	2	49	21	57
20.7	3	LNT + DIEP	SIEA	2	28	17	39
20.7	2	LNT	SCIP	2	27	7	74
21.0	3	LNT	SCIA	2	27	17	37
21.2	2	LNT	SCIP	1	20	9	55
21.7	2	LNT	SCIP	2	21	7	67
21.8	4	LNT + FALD	SCIA	2	12	1	92
21.8	3	LNT	SCIP	3	31	13	58
22.0	3	LNT	SIEA	2	40	27	33
22.2	4	LNT	SIEA	2	30	8	73
22.4	5	LNT	SIEA	2	35	14	60
22.8	3	LNT	SIEA	3	28	2	93
23.2	4	LNT	SCIA	3	37	11	70
23.2	2	LNT + DIEP	SIEA	3	57	26	54
23.4	5	LNT + DIEP	SIEA	3	13	5	62
23.8	4	LNT + DIEP	SIEA	3	32	14	56
24.2	2	LNT	SIEA	3	27	14	48
24.4	3	LNT	SCIA	3	21	11	48
24.5	3	LNT + DIEP	SIEA	3	41	25	39
19.8	2.8						49.6

Abbreviations. LNs: lymph nodes, preop VD%: preoperative Volume Difference in %, postop VD%: postoperative Volume Difference in %, VDR: Volume Differential Reduction, LNT: Vascularized Lymph Node Transfer, DIEP: deep inferior epigastric perforator flap, FATALD: fat augmented latissimus dorsi flap, SIEA: superficial inferior epigastric artery, SCIA: superficial circumflex iliac artery, SCIP: superficial circumflex iliac perforator.

**Table 2 cancers-13-06198-t002:** Group B: Large lymph node flaps (>25 cm^2^).

Flap Size (cm^2^)	Number of LNs	Harvested Flap	Vascular Pedicle	Lymphedema Stage	Preop VD%	Postop VD%	VDR
25.2	3	LNT	SCIP	1	38	16	58
25.5	4	LNT	SCIA	3	67	24	64
25.6	4	LNT + FALD	SCIP	1	12	3	75
26.6	4	LNT + DIEP	SIEA	1	30	13	57
27.2	3	LNT + FALD	SCIA	2	27	14	48
27.3	3	LNT	SIEA	3	63	26	59
27.3	2	LNT	SIEA	3	41	13	68
28.1	3	LNT	SIEA	1	16	4	75
28.4	3	LNT + FALD	SCIP	3	33	16	52
29.9	5	LNT + DIEP	SCIA	2	33	10	70
31.3	4	LNT + FALD	SCIA	2	18	4	78
31.6	3	LNT	SIEA	2	26	11	58
33.8	4	LNT	SCIA	2	29	18	38
34.5	4	LNT	SCIP	2	43	20	53
34.6	4	LNT + DIEP	SIEA	2	30	12	60
36.9	4	LNT + DIEP	SIEA	2	28	15	46
36.9	4	LNT	SCIA	2	44	19	57
37.0	3	LNT	SCIA	2	41	13	68
37.2	4	LNT + DIEP	SCIA	2	38	8	79
39.1	5	LNT + DIEP	SCIA	2	36	14	61
39.1	5	LNT + DIEP	SCIA	2	35	13	63
39.6	4	LNT	SCIA	3	39	13	67
39.6	4	LNT	SIEA	3	30	10	67
39.8	4	LNT	SCIP	2	35	17	51
39.8	4	LNT	SCIA	2	61	29	52
39.8	3	LNT	SIEA	2	31	13	58
40.6	4	LNT	SCIP	2	34	10	71
41.0	4	LNT + DIEP	SIEA	3	30	14	53
41.9	4	LNT + DIEP	SIEA	3	48	20	69
42.3	4	LNT + DIEP	SCIP	3	28	8	72
44.6	4	LNT + DIEP	SCIP	3	40	14	65
47.5	4	LNT + DIEP	SIEA	1	36	12	67
35.0	3.8						61.8

Abbreviations. LNs: lymph nodes, preop VD%: preoperative Volume Difference in %, postop VD%: postoperative Volume Difference in %, VDR: Volume Differential Reduction, LNT: Vascularized Lymph Node Transfer, DIEP: deep inferior epigastric perforator flap, FATALD: fat augmented latissimus dorsi flap, SIEA: superficial inferior epigastric artery, SCIA: superficial circumflex iliac artery, SCIP: superficial circumflex iliac perforator.

## Data Availability

Data are contained within the article.

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
