# Peer review of "Long-Term Outcomes of Lymph Node Transfer in Secondary Lymphedema and Its Correlation with Flap Characteristics"

_cancers, 2021, doi:10.3390/cancers13246198_

Round 1
Reviewer 1 Report
This is a very interesting, nicely designed and reported study.
It seems that the main finding of the study (and the message that authors should prioritize) is that transfer of more LN = better outcomes. This should be very highlighted and stressed through the manuscript, from the title, to the abstract, to an actual hypothesis in the introduction (and any summary of relevant previous literature on # of LN transferred and outcomes or "functional capacity" of transferred LN), methods, results (authors should run an analysis between # of LN transferred and functional outcome -not just flap size-), discussion (has this been reported before? are your findings similar or different and why?), and conclusions.
All other data and information is OK but seems to be secondary to the main finding mentioned above, and -at times- distracting: ok to keep but possibly reduce emphasis/space to favor main message above.
On the same line, it would be interesting to further investigate the main finding. Authors correlate flap size to VDR: what about LN number and VDR?
Have you looked whether different flaps have different average LN numbers? What about other patient characteristics (age, BMI, sex, etc)? Even more interestingly: have you identified any "threshold"of flap size/LN # that is sufficient to achieve good outcomes or the correlation is linear and "the more the better"? For example, from Fig 3 it seems that although a correlation is observed, several "smaller" flaps under 30cm2 still achieve excellent outcomes (actually the best performers are two "small" flaps). Any insight?
Finally, remember to include a paragraph in your discussion to report the limitations of your study and what it could be improved in the future.
Author Response
Dear Madam/Sir (Reviewer 1),
thank you for reviewing our manuscript and providing some suggestions to improve our study.
We considered all of your comments and made the following changes:
A hypothesis sentence at the introduction was added,
an analysis between number of LNs and outcomes also added at the results and discussion,
a comment of a previous study with similar foundings also added at the discussion,
as well as paragraph with the limitations of the study at the end of the discussion.
We hope that the revised manuscript will be suitable for publication in "Cancers".
Thank you again
Reviewer 2 Report
This study addresses an important problem in patients with lymphedema after breast cancer surgery. The retrospective nature of the study is the primary limitation, with inherent bias that is difficult to assess. Nonetheless, I think it does provide important information to the available literature related to flap size after lymph node transfer in patients with lymphedema after breast cancer surgery.
I think that the authors should acknowledge to a greater degree the weaknesses of a retrospective review. If they do this, I think this is an important study.
English language and style are fine/minor spell check required
Author Response
Dear colleague,
thank you for your kind comments.
We added at the discussion a paragraph, reporting the limitations of the study.
Thank you again for your valuable suggestions.